# Metabolic Escape Routes of Cancer Stem Cells and Therapeutic Opportunities

**DOI:** 10.3390/cancers12061436

**Published:** 2020-05-31

**Authors:** Alice Turdo, Gaetana Porcelli, Caterina D’Accardo, Simone Di Franco, Francesco Verona, Stefano Forte, Dario Giuffrida, Lorenzo Memeo, Matilde Todaro, Giorgio Stassi

**Affiliations:** 1Department of Health Promotion, Mother and Child Care, Internal Medicine and Medical Specialties (PROMISE), University of Palermo, 90127 Palermo, Italy; alice.turdo@unipa.it (A.T.); daccardocaterina@gmail.com (C.D.); matilde.todaro@unipa.it (M.T.); 2Department of Surgical, Oncological and Stomatological Sciences (DICHIRONS), University of Palermo, 90127 Palermo, Italy; tania.porcelli93@gmail.com (G.P.); simone.difranco@unipa.it (S.D.F.); francescoverona91@hotmail.it (F.V.); 3Department of Experimental Oncology, Mediterranean Institute of Oncology (IOM), 95029 Catania, Italy; stefano.forte@grupposamed.com (S.F.); dario.giuffrida@grupposamed.com (D.G.); lorenzo.memeo@grupposamed.com (L.M.)

**Keywords:** cancer stem cells, cancer metabolism, metabolic reprogramming, glycolysis, OXPHOS, lipid metabolism, tumor microenvironment, metabolism-based anticancer drugs

## Abstract

Although improvement in early diagnosis and treatment ameliorated life expectancy of cancer patients, metastatic disease still lacks effective therapeutic approaches. Resistance to anticancer therapies stems from the refractoriness of a subpopulation of cancer cells—termed cancer stem cells (CSCs)—which is endowed with tumor initiation and metastasis formation potential. CSCs are heterogeneous and diverge by phenotypic, functional and metabolic perspectives. Intrinsic as well as extrinsic stimuli dictated by the tumor microenvironment (TME)have critical roles in determining cell metabolic reprogramming from glycolytic toward an oxidative phenotype and vice versa, allowing cancer cells to thrive in adverse milieus. Crosstalk between cancer cells and the surrounding microenvironment occurs through the interchange of metabolites, miRNAs and exosomes that drive cancer cells metabolic adaptation. Herein, we identify the metabolic nodes of CSCs and discuss the latest advances in targeting metabolic demands of both CSCs and stromal cells with the scope of improving current therapies and preventing cancer progression.

## 1. Introduction

Despite significant advances in cancer prevention and treatment, metastatic disease is mostly incurable and resistant to common therapeutics.

The development of cancer depends on a small pool of tumor cells owing a phenotype comparable to normal adult stem cells, called cancer stem cells (CSCs), which are characterized by self-renewal and multilineage differentiation capabilities, as well as a great ability to initiate and promote tumorigenesis, metastasis formation and anticancer therapy resistance [1]. CSC features are also dictated by paracrine interactions occurring between tumor cells and their neighboring tumor microenvironment (TME), mainly composed by adipocytes, cancer-associated fibroblasts (CAFs), endothelial and immune cells. CAFs constitute the predominant cellular portion of TME thereby encouraging tumor expansion and progression by providing cytokines, growth factors, metabolites and extracellular matrix remodeling proteins [2,3].

Altered metabolism is a common feature of cancer. Normally, non-cancerous cells mainly catabolize glucose by oxidative phosphorylation (OXPHOS) in the mitochondria to produce ATP [4]. However, proliferating cancer cells metabolize significant amounts of glucose into lactate, even in the presence of normal oxygen, a phenomenon which is known as “Warburg effect” [5]. The preference to utilize glycolysis offers several advantages to cancer cells including adaptation in hypoxic and acidic environments (lactate production), which help cancer cells to rapidly proliferate and invade and surrounding tissues [6].

However, cancer cells are not characterized by a unique metabolic profile. Both intrinsic and extrinsic factors, such as a nutrient-poor microenvironment, have critical roles in determining cell metabolic phenotypes.

Indeed, it is well-established that CSCs can make use of their fully proficient mitochondrial respiration capacity, to face scarce nutrients supply or a hostile microenvironment, with a process named “reverse Warburg effect” [7].

CSCs rely on the release of nutrients, such as amino acids and fatty acids (FAs), by TME stromal cells, to accomplish the biosynthetic and energy requirements necessary to proliferate and metastasize even in a low oxygen microenvironment. One conceivable mechanism of interchange between neoplastic and stromal cells is represented by the release of miRNAs and exosomes, small particles containing nucleic acids (DNA, mRNA and miRNA), metabolites and inflammatory factors [8]. Moreover, exosomes and their cargo are in charge for the establishment of the pre-metastatic niche and, importantly, may contribute to cancer treatment failure [9,10].

Increasing the knowledge about the heterogeneity of CSC’s metabolic demands may be an essential tool for designing novel personalized therapeutic approaches aimed at enhancing the efficacy of treatments by restricting tumor metabolic adaptation.

Here, we will focus on the molecular mechanisms associated with both establishment and maintenance of CSC metabolic phenotypes listed in Table 1, and the biological processes that influence cell metabolic choices including hypoxia, reactive oxygen species (ROS), epithelial tomesenchymal transition (EMT) and the administration of anticancer therapies.

## 2. How Foodie Can Be Cancer Stem Cells? Glucose as the Main Metabolic Fuel

One of the most appointed metabolic hallmark of cancer cells was postulated by Otto Warburg in the early twentieth century [11]. The ‘Warburg effect’ foresees the conversion of pyruvate to lactate, from glucose, under aerobic conditions with a consequent predominance of glycolysis on OXPHOS [12]. This metabolic reprogramming has been initially ascribed to defects on mitochondrial function [13]. According to Warburg an injury on mitochondria respiration is at the base of the oncogenic transformation and the shift from OXPHOS toward glycolysis is persistent in the cancer cell progeny [13]. Recently, in a model of colorectal cancer it has been proved that tumor initiation is sustained by an enhancement of the glycolytic program in a condition in which the mitochondrial pyruvate carrier is inactivated, causing low pyruvate import, necessary for oxidative metabolism, onto the mitochondria. In vivo loss of mitochondrial pyruvate carrier 1 increased high-grade adenoma formation and it was fundamental for the subsistence of tumor initiating cells at the base of colon crypt [14].

Stem and cancer cells prefer glycolysis because it perfectly meets their metabolic requirements [15,16]. Notwithstanding glycolysis is an inefficient process with respect to ATP production, cancer cells uptake high quantity of glucose that enters glycolysis pathway and generates a comparable number of ATP molecules as OXPHOS [17]. Glycolysis is fundamental for cell bioenergetics as well as for the generation of metabolic intermediates used to produce NADPH, lipid or glycogen molecules. For instance, ribose-5-phosphate groups derived from glucose-6-phosphate are used for nucleotide biosynthesis [18]. In this context an enhanced nucleotide production, as it is the case in cancer cells, leads to a decrease in the NAD+/NADH ratio, which in turn encourages the conversion of glucose to lactate [19].

Glycolysis counteracts the formation of reactive oxygen species (ROS) in proliferating cells [28]. Luo et al. demonstrated that the inhibition of glycolysis by 2-deoxyglucose (2-DG) induced the transition of mesenchymal breast CSCs harboring low levels of ROS toward a ROS^high^ phenotype of epithelial breast CSCs, which are sensitive to auranofin and L-buthionine-sulfoximine (BSO), two inhibitors of the antioxidant pathways thioredoxin (TXN) and glutathione (GSH), respectively [27,29]. The latter study clearly indicates that despite glycolysis plays a central metabolic role, cancer cells modulate their metabolic phenotype in response to metabolic stress, including the possibility to switch to OXPHOS and vice versa if needed [30].

Indeed, pancreatic CSCs harboring high level of the peroxisome proliferator-activated receptor gamma co-activator 1-alpha (PGC-1α) rely on OXPHOS and diverge from the glycolytic phenotype of their differentiated counterpart. Mitochondrial inhibition by metformin causes the selection of a resistant subtype of MYC-expressing pancreatic CSCs showing an intermediate glycolytic/oxidative metabolic phenotype [24].

## 3. The Mitostemness: Fiction or Reality?

A conspicuous entity of studies shows that cancer cells are characterized by high energy demand. According to the “Warburg effect” cancer cell sustenance is solely glycolytic.Nowadays we know that although the Warburg phenotype represents an undisputed feature of many cancer cells, most tumor cells possess intact tricarboxylic acid (TCA) cycle and OXPHOS [31,32].

The oxidative metabolism allows CSCs to have selective advantages, greater resistance to the inhibition of glycolysis, enhanced degree of independence from the supply of macronutrients and also a more efficient source for energy production [20,33]. In order to emphasize the central role of mitochondria in the self-renewal of CSCs and in their resistance to differentiation, two years ago it was coined the term “mitostemness” [34,35].

Several studies have highlighted that CSCs are endowed with a plastic metabolic phenotype, both glycolytic [36] and oxidative through the OXPHOS [37]. The latter process is mediated by electron transport chain (ETC) and involves supply of reducing equivalents from TCA cycle or fatty acid oxidation (FAO) for ATP and NADPH generation.

Janiszewska M. et al. demonstrated that the subpopulation of tumor-initiating gliomaspheres depends on OXPHOS for energy production. In particular, the oncofetal insulin-like growth factor 2 mRNA-binding protein 2, which regulates the protein synthesis and contributes to assembly of the subunits of ETC, sustained glioblastoma cell clonogenicity [21].

Mitochondrial metabolism is fundamental in CSCs because it constitutes a source of intermediate metabolites such as acetyl-coA, NAD+, alpha-ketoglutarate, succinate, FAD, S-adenosylmethionine. The variation in the concentration of these intermediate metabolites could lead to important phenotypic changes because they can be used by enzymes capable of modifying the state of the chromatin [38] and thus the transcriptomic profile of cancer cells. There are various ways to modify chromatin, for example, acetyl-coA is used to transfer acetyl group to lysine residues of histones [39] and S-adenosylmethionine is a donor of methyl group for the methylation of DNA and histones [40]. Histone acetylation, mediated by histone acetyl transferases, increases gene expression [41]. The proteins involved in the cytosine methylation of CpG dinucleotides are the DNA methyl transferases DNMT1, DNMT3A and DNMT3B and they have been identified as factors driving the CSCs formation and maintenance [42]. Histone methylation occurs predominantly on lysine (K) and arginine (R) residues. Such modifications are commonly associated with gene activation or repression, depending on the target histone modification [43]. Interestingly, the enhancer of zeste homolog 2 (EZH2), which is the catalytic subunit of polycomb repressive complex 2, has been demonstrated to downregulate gene transcription through trimethylation of histone H3 on lysine 27 [44], while it upregulates NOTCH expression, being crucial in the expansion of the stem pool in breast cancer [45].

Given the discovery of functional mitochondria in CSCs, it was hypothesized that the capability to rapid switch from glycolytic metabolism toward oxidative metabolism [46] and vice versa, could represent an escape mechanism to standard anticancer therapies. Therefore, using drugs that selectively target mitochondria could make CSCs more sensitive to standard therapies.

PGC-1α is a stress sensor, activated following the presence of limited amount of nutrients, oxidative stress and chemotherapy, to increase mitochondrial biogenesis and therefore all mitochondrial activities (OXPHOS, FAO and detoxification ROS) [47]. The XCT790 is an inhibitor of estrogen-related receptor α, the latter being a transcription factor coupled with PGC-1α. In this context, De Luca et al. have shown that the use of XCT790 reduces the proliferation of CSCs in breast cancer [48].

One of the most studied ETC inhibitor to target CSCs is the antidiabetic drug metformin, which targets the complex I in ETC in mitochondria. Metformin acts as driver of apoptosis in CD133+ pancreatic ductal adenocarcinomas cells and CD44^high^CD24^low^breast cancer spheres [24,49]. Promising clinical data havebeen reported for the use of metformin in breast, endometrial, prostate and pancreatic cancers (NCT01266486; NCT02755844; NCT01620593; NCT02978547) [50,51]. For instance, a retrospective study conducted on 445 patients affected by neuroendocrine pancreatic tumors revealed that progression free survival was doubled in diabetic patients receiving metformin than patients without diabetes (median progression free survival, 32.0 months versus 15.1 months) [51].

Another important mitochondrial target is represented by mitochondrial ribosomes, which are the site of protein synthesis. The homology existing between mammalian mitochondrial ribosomes and prokaryotic ribosomes is well established, for this reason it is possible to use a wide variety of antibiotics already approved by the FDA [52]. Ribosomes are targeted by the two main families of antibiotics, tetracycline and macrolide [53,54]. Among the wide range of antibiotics, tested by Lamb et al. doxycycline was found to inhibit mammosphere formation in breast cancer and many different tumor types [52]. Therefore, this therapeutic approach could be applicable to many cancer types.

Notwithstanding inside the cell there are various production sites of ROS, mitochondrial ETC represents their primary endogenous source [55]. The intracellular increase of ROS may result from the activation of oncogenes, inactivation of tumor suppressor genes, enhanced oxidative metabolism and mitochondrial dysfunction [56]. The accumulation of ROS causes damage to proteins, lipids and DNA and therefore irrecoverable damage to the cell [27]. Notably, cells have protection mechanisms from oxidative stress, including enzymatic antioxidants such as superoxide dismutase (SOD), catalases, TXN, peroxiredoxins, glutathione (GSH) peroxidases, p38-MAPK and sirtuins [57,58,59] and non-enzymatic molecules such as GSH, vitamin C (ascorbate), vitamin E (tocopherols) and polyphenols [60]. Interestingly, the administration of SOD-1 inhibitor, disulfiram, strongly counteracted breast CSCs expansion by modulating MAPK, NF-kB and STAT3 pathways [61,62].

Although it has been shown that there is an increase in mitochondrial functionality in CSCs compared to other cancer cells [48], it appears that there are low concentrations of ROS within CSCs [63,64] as there is a high production of antioxidant agents that maintain the disposal of ROS, protecting CSCs from oxidative damage [65]. The excessive increase in ROS levels reduced clonogenicity and mediated death of CSCs [66]. Moreover, the use of paraquat, an inducer of ROS, increased ROS production and significantly reduced tumor spheroid formation of CSCs [16].

Albeit several authors have shown that CSCs harbor high ROS levels, further studies clarified their role in cancer onset and progression. In particular, ROS facilitates CSCs self-renewal, expansion, invasiveness [67,68,69] and potentiated tumor progression [70,71].

Myant et al. highlighted the connection between Rac1, NF-kB and ROS in CSCs, in which ROS act as connecting molecules between Rac1 and NF-kB. Rac1 is required for NF-kB activation and initiation of colon tumorigenesis. Rac1 is also part of a protein complex containing NADPH oxidase, which generates superoxide, a major constituent of the pool of intracellular ROS. On the other hand, NF-kB is a redox-sensitive transcription factor that is activated by increased levels of ROS. In this study, the authors showed that increased NADPH oxidase activity, ROS intracellular concentration and NF-kB signaling in the APC-inactivated intestinal cells are all dependent on Rac1 expression [68,72].

Within tumor bulk, CSCs represent the subpopulation with the highest degree of adaptability to redox stress. The GSH system, adopted by CSCs, is the major antioxidant defense mechanism that reduces the intracellular levels of ROS. Gln-derived glutamate (Glu) is a necessary substrate for the synthesis of GSH and an exchange ion for the cellular import of cystine, which cooperates with Glu for the production of GSH, through the CD44v-xCT transporter [73]. Cells showing stem-like features strongly rely on mitochondrial OXPHOS sustained by glutamine (Gln) catabolism [23]. Of note, cancer cells are metabolically heterogeneous, since they can also retain the OXPHOS capacity and replace glucose with Gln or fatty acids (FAs), as energy supply. Despite being a non-essential amino acid, Gln is considered as a ‘conditionally essential’ amino acid since its de novo synthesis does not satisfy the demands of cancer cell, which become reliant on its exogenous uptake. Gln is also required for TCA cycle intermediate replenishment, protein translation and the biosynthesis of amino acids and nucleotides [74].

Interestingly, metformin resistance can be mediated by Gln compensation. In human cancer organoids and in in vivo model it was described that the administration of metformin and a glutaminase inhibitor (GLSi), which blocks Gln consumption, efficiently overcome metformin resistance [75]. Glutaminase activity also regulates head and neck cancer metabolism though the expression of the stemness marker ALDH [25].

It has been shown that survival of lapatinib-resistant breast cancer cells is supported by Gln metabolism [76], whose inhibition could be exploited for sensitizing cancer cells to standard therapies. Notably, GLSi have been recently launched to Phase II in combination with paclitaxel for the treatment of triple negative breast cancer patients (NCT03057600).

Recently, different clinical studies demonstrated that leukemia stem cells are greedy of amino acids and possess enhanced catabolic activity [26]. Some amino acids, such as methionine, influence the epigenetic state of cancer cells and fuel tumor initiation [77]. In addition, glycine decarboxylase, belonging to the serine–glycine pathway, is the metabolic driving force of tumor-initiating cells in non-small cell lung cancer [22]. Metabolic targeting of amino acids resulted often in neoplasia regression [77]. However, phenomena of cancer metabolic compensation, mediated by FA oxidation, have been observed [26] (Figure 1).

## 4. Fatty Acids as a Metabolic Reservoir for Cancer Cell Plasticity

As we have already extensively discussed in this review, the ability to adjust cell metabolism is crucial in cancer cells, because it supports the neoplastic proliferation and survival during the different steps of cancer progression [78]. Given the high plasticity of CSCs, this phenomenon is highly pronounced in this cell subset, in particular regarding FA metabolism, which is fundamental to sustain their boosted growth, division and survival during cancer progression. FAs play an important role in different aspects of cancer cell life, being crucial for cell membranes assembly, acting as structural components of the extracellular matrix, secondary messengers and energy source [79]. FAs abundance is regulated by two different processes, de novo biosynthesis mediated by nutrients as glucose and amino acids, and exogenous uptake thanks to the presence of specific transporters, both mechanisms are finely regulated at different levels during cancer progression by the dynamic TME components and dysregulation of oncogenic signaling pathways (reviewed in [80]). Importantly, in 1984, Ookhtens and colleagues demonstrated in vivo that most of the esterified FAs in tumors were derived from de novobiosynthesis [81], as confirmed few years later by Kuhajdaet al., who showed for the first time that tumor cells (breast cancer in this study) over-express fatty acid synthase (FASN) [82]. These findings were also demonstrated in glioma, breast and pancreatic CSCs, where FASN expression dictated the acquisition of stemness features [83,84,85,86].

Lipid droplets (LDs) are dynamic and multifunctional cytoplasmic organelles involved in the storage of FAs, which are fundamental to avoid lipotoxicity when cells start accumulating an excess of FAs, in particular under hypoxic conditions that induce an upregulation of FAs uptake, but also to provide an important source of ATP and NADH under stress. The generation of energy from LDs is achieved by β-oxidation of stored lipids that is sufficient to provide ATP during the metastatic cascade and essential for production of NADPH useful as detox agent against ROS [87,88]. The role of LDs biogenesis and their regulated mechanisms in cancer cells has been widely described over the last decade, and observed in all the phases of cancer development, including initiation, promotion and progression [89,90] in particular in CSCs from different organs, including colon cancer [91,92]. These findings suggest that LD biogenesis could be considered a promising target for designing innovative antitumor therapeutic strategies [93,94].

In addition to de novo synthesis, FAs can be collected from exogenous environment through the expression of specialized transporters that facilitate the efficient movement across the plasma membrane. The best characterized include CD36, also known as fatty acid translocase (FAT), the fatty acid transport protein family (FATPs) and the fatty acid-binding protein (FABPs), which have been shown to be upregulated in tumors [95,96]. In particular, CD36 has been demonstrated to be crucial for metastatic dissemination of cancer cell in different tumor type, with its high expression being correlated with poor prognosis [87]. Accordingly, targeting of CD36 has shown pre-clinical evidence that its inhibition is sufficient to impair metastases [87]. Interestingly, the expression of CD36 in cancer cells is also able to regulate the metabolic crosstalk with TME by increasing their dependency on exogenous lipid uptake. Lipid uptake from cancer cells becomes crucial mostly under condition of metabolic stress. Indeed, it has been shown that in hypoxia, the glucose to acetyl-CoA flux is downregulated, as well as unsaturation of FA that is driven by stearoyl-CoA desaturase-1 SCD-1, which is an oxygen-consuming enzyme. To compensate this prohibitive microenvironment, during hypoxia cancer cells upregulate FABP4 [97], which is a target of the hypoxia-induced factor 1 alpha (HIF1α) [98]. Notably, FABP4 negatively correlated with progression-free survival [99].

Adipose tissue cells establish symbiotic relationship with cancer cells. Interestingly, some cancers show preferential homing to adipose tissue [100]. Adipocytes sustain cancer cell growth by activating lipolytic processes with consequent release of free FAs that are up taken by cancer cells over-expressing FABP4, which in turn release exosomes containing pro-lipolytic factors (i.e., miR-144 and miR-126) [101,102]. We can distinguish two different populations of cancer adipocytes, the “peritumoral adipocytes” that do not penetrate tumor tissue (surrounding tumor tissue, tumor-educated adipose cells can be also found distant from tumor growth [103]), and the “metastasis-associated adipocytes”, which arise from the transient or prolonged contact between metastatic cancer cells and resident adipose cells. The most important mechanism involved in cancer cell metabolic reprogramming, is driven by cancer cell-driven lipolysis in adipose cells [104]. Indeed, it has been demonstrated that following prolonged exposure of cancer cells to adipose cells, the latter start losing their lipid content (responsible for tumor cachexia), gaining fibroblast-like phenotype, which further promotes the metastatic behavior of cancer cells. Recently, Ye et al. demonstrated that leukemic stem cells use gonadal adipose tissue as niche to support their growth and to protect themselves from chemotherapy, by inducing an inflammatory microenvironment and adipose tissue lipolysis [105]. Interestingly, the most beneficial effect was gained by leukemic stem cells over-expressing the FA transporter CD36 [105].

Another important source of heterogeneity in the metabolic signature of cancer cells is due to the genetic background since several metabolic processes are directly regulated by constitutive activation of oncogenes and deactivation of tumor suppressor genes [106]. In particular, oncogenes as *RAS* and *PI3K* are usually associated with glycolysis over OXPHOS, while tumor suppressor genes, as p53, have opposite effects [107].

One of the most important oncogenes in cancers is *RAS*. Indeed, constitutive activation of KRAS is frequent in colon and non-small cell lung cancer [108]. *KRAS* by activating ERK1/2 reprogramscancer cell lipid metabolism, leading to increased synthesis of glycerophospholipids and FAs [109,110]. G12V mutated *KRAS* is also able to boost de novo lipogenesis [111].

PI3K is one of the most frequently dysregulated pathways in cancers. Constitutive activation of this signaling regulates de novo lipid biosynthesis by increasing acetyl-CoA synthesis [112], and promoting NADPH production to fuel lipogenesis [113]. Several metabolic pathways are also regulated by mTOR, which is strictly linked to dysregulation of PI3K/Akt pathway. mTOR can affect cancer cell lipid metabolism by activating OXPHOS that in turn promote lipogenesis [114], as recently demonstrated by the evidence of downregulation of lipogenic enzymes, including FASN, acetyl-CoA carboxylase 1 and ATP citrate lyase, following treatment with mTOR inhibitor or by raptor genetic knockdown [111,115]. PI3K can be activated by stimulation of growth factor receptor tyrosine kinases, such as HER2. HER2 positive tumors are characterized by enhanced de novo lipogenesis, which contributes to the aggressiveness of these tumors [116]. The overexpression of HER2 is sufficient to prompt a FASN-dependent lipogenic phenotype in non-transformed epithelial cells that recapitulate the cancer cell metabolism. Conversely, HER2 inhibition—or de novo lipogenesis—inhibit oncogenic potential and induce apoptosis [117].

Among the others, p53 can directly or indirectly regulate lipid metabolism-related gene expression [118]. p53 downregulates FAs synthesis by inhibiting the pentose phosphate pathway and downregulating SREBP1 [119]. On the other hand, p53 is able to increase FAs oxidation and prevent lipid accumulation, by inhibiting the pyruvate dehydrogenase kinase at transcriptional level, thus leading to increase in pyruvate dehydrogenase and to the conversion of pyruvate to acetyl-CoA [120,121].

The complex intra-tumor regulation of metabolic activities is further enriched at cellular level, as best represented by breast cancer, which includes multiple cellular subtypes characterized by specific hormone/growth factor receptor and genetic profile. A recent study has highlighted how triple-negative breast cancers over-express genes involved in exogenous lipid uptake (high FABP5 and FABP7 expression), while the receptor-positive breast cancers are associated with de novo lipogenesis [122]. For these reasons, a better characterization of lipid-associated signatures in cancers could help to guide therapeutic interventions.

In additionto fuel the boosted proliferation, aberrant FA metabolism protect cancer cell from surrounding stress of different nature. It has been demonstrated that an excess of saturated FAs is responsible of mitochondrial/ER stress and dysfunction [123]. To prevent accumulation of saturated FAs, cancer cells overexpress SCD1 and SCD5, which convert saturated into monounsaturated FAs [124]. To cope with hypoxic microenvironment that often characterizes tumor progression, and which inhibits the activity of SCD enzymes, cancer cells upregulate the uptake of exogenous unsaturated FAs in order to maintain lipid homeostasis [123]. Importantly, to increase FAs uptake, it has been demonstrated that cancer cells over-express lipoprotein–lipase, which are involved in lipolysis of extracellular TG-rich lipoproteins, thus facilitating the hypoxia-induced FA uptake directly from TME [125]. However, hypoxia-induced regulation of FAs is highly cell-specific [126]. The major difference among the available studies is due to the cell culture conditions, and in particular, by the abundance of nutrients in cell culture media. Indeed, it is now clear that hypoxia or serum starving alone induce different metabolic phenotype in cancer cells, compared to their combination. Importantly, hypoxic regions are characterized by both oxygen and nutrients deprivation. In such condition, cancer cells are dependent on Gln and acetate as carbon source for the production of acetyl-CoA [127,128]. In line with these findings, when both oxygen and serum are limited, cancer cells upregulate nuclear acyl-coenzyme A synthetase that plays a dual role, aiding the use of extracellular acetate as carbon source and maintaining high histone acetylation levels, to prevent induction of apoptosis and promoting cancer cell growth [129].

Another important consequence of hypoxia microenvironment is represented by the excess in ROS production. This phenomenon results from FA oxidation process, activated by patatin like phospholipase domain containing 2 (PNPLA2) and mitochondrial electron leakage, which both lead to oxidative stress. To counteract this phenomenon, cancer cell downregulate PNPLA2 through the hypoxia-inducible protein 2, to promotes cancer cell survival [130,131]. The resistance to oxidative stress in hypoxic condition is also driven by a deregulated membrane lipid saturation, with increased saturated and monounsaturated FAs, which are less susceptible to peroxidation [132].

Dysregulation of membrane lipids saturation, as well as cholesterol content, could also affect cell membrane fluidity and dynamics, both mechanisms associated with the acquisition of a mesenchymal phenotype, which is crucial in metastatic cancer cells. Indeed, cholesterol synthesis is finely regulated during metastatic dissemination of cancer cells. Ehmsen, et al. recently demonstrated that elevated cholesterol biosynthesis identifies the subpopulation of breast CSCs. The administration of simvastatin, by inhibiting the 3-hydroxy-3-methyl-glutaryl-coenzyme A reductase, hampered mammosphere formation [133].

All these pre-clinical findings regarding the role of FA metabolism in cancer progression arouse clinical interest for development of innovative antitumor strategies targeting FA metabolism, each one directed against specific phases of lipid metabolism: de novo synthesis, modification, uptake, activation, storage, mobilization and degradation [134].

Most of the enzymes involved in FA metabolism are regulated by sterol regulatory element binding protein-1SREBP-1 at transcriptional level. However, despite targeting SREBP-1 could potentially represent a promising strategy to regulate lipid metabolism in cancer cells, transcription factors are mostly considered undruggable. For this reason, several pre-clinical studies are now considering the idea to target its protein partner, SREBP-cleavage activating protein, which drive SREBP-1 to the Golgi for activation, by treating cancer cells with fatostatin and betulin [135,136]. Another important druggable player in FA biosynthesis is FASN. Although several pre-clinical studies showed efficacy in inhibiting cancer growth, the pronounced side effects, mostly on neuronal stem cells, impeded its use in clinical settings [137,138]. FA modification is an important step in lipid metabolism. Among the enzymes driving this step we find the elongases (seven members, ELOVL1-7) and desaturases (two members, SCD-1 and SCD-5). Importantly, our group has recently shown that inhibition of SCD-1, using the betulinic acid, in vitro is sufficient to efficiently kill colon cancer cells, in particular the CSC subset, leading to mitochondria-dependent cell death [139,140]. Similarly, the SCD-1 inhibitor, CAY10566, inhibits glioblastoma stem cells expansion and in vivo tumor growth [141]. As previously mentioned, one of the most important transporters of FAs is CD36. Targeting this membrane protein has been demonstrated to impair metastases in breast cancer and melanoma, by inhibiting the metastasis-initiating cell compartment [87].

## 5. The Role of miRNAs in Cancer Stem Cells Metabolism

Recent studies shed light on the role of microRNAs (miRNAs) in regulating stemness features of cancer cells, such as self-renewal, differentiation, metabolic reprogramming, metastasis formation and anticancer therapy resistance, associated with the pathogenesis of various types of human cancer [142].

miRNAs are single strand short non-coding RNA molecules (21–23 nucleotides) transcribed as precursor molecules, which are subsequently cleaved by the endoribonucleases Drosha and Dicer. Mature miRNAs are bound by a member of the Argonaute(AGO) protein family to form the RNA- induced silencing complex (RISC) in a process termed RISC loading. The miRNA guides RISC to complementary sequences located mainly in the region of its target mRNAs, silencing gene expression [143]. Following nuclear transcription, miRNAs are processed within the "multi-vesicular bodies", that consist of phospholipid membranes and lipoprotein microvesicles of endocytic origin, known as exosomes, of about 30–100 nm. After the binding of bodies to the plasma membrane, miRNAs are released into the bloodstream [144].

miRNAs have the role of coordinating gene expression programs at the base of physiological and pathologic cellular processes, including cancer [145]. Interestingly, miRNAs can act as both oncosuppressors, inhibiting proliferation and oncogenes, promoting tumor initiation, growth and metastasis formation [146]. Each miRNA can bind and regulate the expression of multiple coding or non-coding mRNAs. Therefore, the aberrant expression of a single miRNA can deleteriously influence the translation of multiple genes within a cell, leading to profound phenotypic changes [146].

To better understand the interaction between miRNAs and CSCs, recent studies evaluated how miRNAs can play a key role in maintaining and regulating the functioning of CSCs by targeting various oncogenic signaling pathways, such as Notch, WNT/β-catenin, JAK/STAT, PI3K/AKT and NF-kB [142].

Some miRNAs, such as miR-145, miR-200c, miR-494, miR-195-5p, miR-34, miR-519d, miR-128, miR-99a inhibit CSCs expansion by inhibiting of ADAM, BMI, Notch, caspases and mTOR, respectively, while others, such as miR-19, miR-501-5p, miR-21, miR221/222, miR-483-5p, miR-196b-5p, miR-494-3p stimulate stemness through the activation of WNT/β-catenin, PTEN, cyclin D1 and MMP-2, STAT3 and Notch 1 pathways [142].

The deregulation of Drosha and Dicer has been observed in different types of cancer [147,148]. For instance, the downregulation of Dicer led to decreased miR-130b and tumorigenesis [149]. In addition to Drosha and Dicer, other enzymes involved in the miRNAs biogenesis pathway, are the trans-activation responsive RNA-binding protein (TARBP2) and the Argonaute RISC catalytic component 2 (AGO2). In sporadic and hereditary carcinomas, downregulation of TARBP2 protein expression in CSCs was shown to be important for pro-metastasis signaling [150].

A key role in the miRNAs biogenesis pathway is played by Exportin-5 (XPO5) protein. It mediates the nuclear export of miRNAs precursors to the cytoplasm. Genetic mutations of *XPO5* leads to an entrapment and loss of precursor miRNAs in the nucleus promoting tumor initiation through an increase in the expression of stemness-related genes such as *EZH2* and *MYC* [151].

Notably, different miRNAs have been associated with the regulation of cancer cell metabolism [152].

The transport of glucose inside the cells is an important event for the correct cellular functioning. This mechanism is mediated by tissue-specific glucose transporters known as GLUTs, which are directly or indirectly targeted by several miRNAs. Presumably, deregulation of GLUTs can increase glucose uptake, satisfying the high glucose requirement and accelerating metabolism in cancer cells. However, the direct links between miRNAs deregulation and glucose transport in cancer are largely unknown therefore further studies are needed. A key enzyme for aerobic glycolysis is the hexokinase 2 (HK2), which is overexpressed in tumors. miR-143 is inversely correlated to the expression of HK2 in the head and neck squamous cell carcinoma and in lung cancer [153]. Likewise, the loss of miR-143, in glioma tissues and glioblastoma stem-like cells, promoted the expression of HK2, resulting in enhanced aerobic glycolysis and inhibition of cell differentiation [154]. The connection between miR-143 and HK2 has been studied also in colorectal cancer, where it has been shown that it downregulates HK2 expression and that its reintroduction leads to a decrease in lactate secretion by impairing the rate of glycolysis [155]. Another miRNA-regulated glycolytic enzyme is phosphofructokinase 1 (PFK1). PFK1 catalyzes the phosphorylation reaction of fructose-6-phosphate to convert it into fructose-1,6-bisphosphate. Yang et al. showed that miR-135 targets PFK1 and inhibits aerobic glycolysis and suppresses tumor growth [146].

Chong et al. studied the role of miRNAs in maintaining the glycolytic metabolism of CSCs [156]. In their study, they demonstrated a critical role of the LIN28B in stemness. In breast cancer cell lines, the inhibition of LIN28B suppresses MYC expression and increased miR-34a-5p levels expression, correlating with inhibition of glucose uptake/lactate production and a better patient’s prognosis. Therefore, blocking of the LIN28B/MYC/miR-34a-5p signaling pathway, by the LIN28B specific inhibitor, causes a dramatic inhibition of tumor growth and metastatic potential in orthotopic immunodeficient mouse models of human breast cancer cells [156].

A class of miRNAs that regulates the choice between self-renewal and differentiation of breast CSC is represented by miR-600. In breast CSCs, the miR-600 silencing caused cancer cell expansion, while its overexpression reduced CSC self-renewal, leading to a decreased in vivo tumorigenicity. Interestingly, the main target of miR-600 is SCD1, an enzyme required to produce active lipid-modified WNT proteins, which are involved in cell fate determination during tissue development and oncogenic processes [157]. Indeed, the authors showed that low levels of miR-600 are correlated with active WNT signaling and a poor prognosis. These findings highlighted that miR-600 is involved in breast cancer cells-fate decisions and influences tumor progression [157].

In addition to glucose, cancer cells need Gln for their growth. This adaptive metabolism by cancer cells appears to provide substrates for an increase in lipogenesis and biosynthesis of nucleic acids which are fundamental for the proliferative phenotype of the cancer cell. Cell transformation has been shown to stimulate glutaminolysis and many cancer cells are closely dependent on this metabolic path [158]. One of the main regulators of glutaminolysis is MYC, which promotes not only cell proliferation, but also the generation of macromolecules and antioxidants necessary for the growth of cancer cells. Interestingly, MYC’s inhibition of miR-23A/B improves mitochondrial expression of glutaminase and Gln metabolism and correlates with the onset of a neoplastic phenotype [158]. Similarly, Wang et al. have demonstrated that MYC, via miR-33b induction, supports glioblastoma CSCs via the activation of mevalonate metabolism [159]. Pyruvate dehydrogenase kinase 4 is target of miR-122, a liver-specific miRNA. mir-122 is able to hamper the glycolytic metabolism in the CD133+ CSC compartment, decreasing spheroids formation and sensitizing to standard anticancer therapy [160].

An increased expression of miR-210-3p correlates with colorectal cancer progression [161]. In fact, the stable overexpression of miR-210 in colorectal cancer spheroid culturesresulted in significantly enhanced CSC self-renewal activity [162]. In colon CSCs miR-210 inhibits mitochondrial TCA cycle activity to enhance lactate production [163]. The lactate stimulation leads to an increased self-renewal capacity of different colon CSC cultures [163].

The rapid proliferation of cancer cells within a tumor mass leads to the formation of hypoxic regions caused by the absence of an efficient vascular network. Cancer cell survival in hypoxia requires the activation of adaptive pathways [164] and the downregulation of miRNA through a reduction of Dicer and Drosha [165,166].

Within tumor is possible that miRNAs processed in the normoxic portion of a tumor diffuse toward hypoxic zones to promote tumor growth and regulate gene expression [149]. In colon CSCs, miR-210 promotes the self-renewal and reprograms cell metabolism toward a prompted glycolytic and lactate yield. In hypoxic conditions HIF1α induces miR-210-3p expression, this determines a reduced oxidative metabolism (TCA cycle and OXPHOS) [161]. miR-210 is an oncogenic miRNA and a target of HIF-1 and HIF-2 [167]. It has been shown that during hypoxia, miR-210 targets the mRNA that encodes the mitochondrial electron transport chain component protein succinate dehydrogenase complex subunit D (SDHD). Downregulation of SDHD results in an increased stabilization of HIF1α and cancer cell survival [168,169]. In the hypoxic microenvironment, miRNAs contribute to metabolic activities, to the maintenance of stemness in cancer cells and to therapy resistance. miRNAs, hypoxia and CSC are part of a complex signaling network that promotes tumor aggressiveness [170].

Several miRNAs target important cancer cell regulatory molecules and are involved in an intricate network of signaling between cancer cells and the TME. In addition to their involvement in direct cell-to-cell signaling, several miRNAs are secreted through micro vesicles or exosomes and affect cancer cell growth and metastasis [165]. Broniszeet al. studied the role of miRNAs in the transformation of normal fibroblasts into CAFs, focusing on PTEN-regulated miR-320. Downregulation of miR-320 and upregulation of one of its direct targets, ETS Proto-Oncogene 2, concomitantly with loss of *PTEN*, is a key event in oncogenic process. In fact, this event leads to increased angiogenesis and tumor formation [171]. miR-320 was found to regulate CAF-secreted proteins, including MMP9, MMP2, lysyl oxidase homolog 2 and elastin microfibril interfacer 2, which are known to enhance tumor metastasis by programming the TME via degradation of extracellular matrices [165]. Hua Y et al., identified an important chemokine in the TME, CCL5, as target of miR-214, the latter being downregulated in CAFs. These data support the idea that miRNAs could alter the TME by changing protein production in CAFs, such as chemokines, sustaining tumor growth [172].

Invasion and metastasis of cancer cells have been associated with phosphoglucose isomerase, thedownregulation of which by mir-200 in breast cancer cells impaired metastasis spread [173]. Indeed, well-known miRNAs that are downregulated in cancer belong to miR-200 family. These miRNAs are involved in many different functions, such as induction of EMT via downregulation of E-cadherin and consequent increases in zinc finger E-Box (ZEB) proteins. It was demonstrated that miR-200 influences angiogenesis indirectly via downregulation of chemokine and interleukin such as CXCL1 and IL-8, which are major players in the TME [174].

Knowledge about the role of miRNAs in development and diseases, particularly in cancer, encouraged the use of miRNAs as tools or as targets for novel therapeutic approaches. For instance, several approaches are based on the use of miRNA mimics to restore the expression of tumor suppressors or antimiR to target oncogenes [169]. miRNAs mimics are synthetic double-stranded small RNA molecules that match the corresponding miRNA sequence. In human disease this therapy is used to replace the lost miRNA expression and their function. In contrast, antimiRs are single stranded and based on first-generation antisense oligonucleotides, which had been designed to target mRNAs or modified with locked nucleic acids.

For example, a mimic of miR-34, whose target is lactate dehydrogenase A, is used like a tumor suppressor in Phase I clinical trials (NCT01829971).

On the other hand, in Phase II clinical trials for thetreatment of hepatitis, it was used antimiRs targeting miR-122 (NCT01200420; NCT01872936; NCT02031133; NCT02508090) [169].

Studies have shown that miRNAs such as let-7, miR-34, miR-451 and miR-200 can sensitize cancer cells to chemotherapy, and mimics of these miRNAs could be rationally combined with chemotherapeutic drugs [175,176,177].

MRX34 was the first miRNA-based therapy undergoing clinical trial for treatment of lymphoma, melanoma, multiple myeloma, liver, lung and renal carcinoma (NCT01829971) [178]. The use of MRX34 in the aggressive *KRAS TP53* mutated non-small cell lung cancer mouse model, led to significant tumor reduction. Moreover, combination treatment with the epidermal growth factor receptor (EGFR) inhibitor, erlotinib and miR-34 mimic and let-7 showed synergistic effects in inhibiting the growth of non-small cell lung cancer cell lines in vitro [177].

A similar example is the MesomiR-1 drug, which reintroduces miR-16, a miRNA that regulates aldolase A in glycolysis process [179,180].

Another therapy approach valuated in clinical trial is the locked oligonucleotide acid-modified inhibitor for miR-155 (MRG-106) (NCT03713320) [181]. MRG-106 replaces both miR-155 and miR-143, that negatively regulates HK2 and counteracts glycolytic phenotype [182].

Given that miR-29 is frequently lost in cancer and has been reported to negatively regulates monocarboxylate transporter 1 (MCT1), a lactate transporter [183,184] it has been tempted a novel approach that involves the use of miR-29 mimic (MRG-201).

In conclusion, although numerous studies have been conducted to understand the role of miRNA in the CSC metabolism, further studies are needed to define their role more precisely.

## 6. The Influence of Microenvironment in Cancer Stem Cells Metabolism

The TME is characterized by a high degree of metabolic heterogeneity and dynamics that involves both cancer and microenvironmental cells. Based on the metabolic activity it is possible to classify tumors in “glycolytic” (lung, liver, colorectal, leukemia) versus “oxidative” (melanoma, glioblastoma) [185]. The complexity of this scenario is even increased by the evidence that tumors originated from the same organ—or characterized by the same genetic background—are not metabolically uniform [186,187]. This intra-tumor metabolic variability is mainly dictated by different access to oxygen and glucose, which occurs in proximity of blood vessels and by the different cell population co-existing in TME. Indeed, cancer and stromal cells can compete and/or cooperate for nutrients [188].

The neoplastic tissue is constituted not only by the tumor cells, but also by the stromal cells; together they constitute the TME. CAFs, a cellular component of the TME, influence tumor growth by supplying nutrients or directly by cell-to-cell communication. In addition, CAFs provide a stromal framework to cancer cells during early growth and development, leading to malignant transformation [149].

The metabolic coupling between cancer and stromal cells, which is responsible for the ‘reverse Warburg effect’, is mediated by an interchange of cytokines, metabolites and miRNAs. Stromal cells allow the creation a permissive soil that sustains cancer cell growth and fuel cancer cell energy by supplying metabolic substrates, including Gln and lactate, which enhance a specific metabolic pathway, such as OXPHOS [105,189,190] (Figure 2).

In nutrient-deprived conditions, CAFs can undergo metabolic adaptation and synthesize Gln that is released in TME for cancer cell consumption, thus fostering tumor growth and metastasis formation. CAF-derived lactate, in turn, promotes OXPHOS in cancer cells, favors Gln uptake and catabolism and is responsible for the resistance to glutaminase inhibitors [191]. Monocarboxylate transporter (MCT) 4 exports lactate from CAFs, while MCT1 allows lactate uptake in CSCs [192].

Another example is the alanine uptake by pancreatic cancer cells. This essential amino acid is released by pancreatic stellate cells by autophagy and fuel TCA cycle in pancreatic ductal adenocarcinoma cells for the production of essential amino acids and lipids [193].

On the contrary, it seems that CSCs prefer glycolysis as source of ATP in a low oxygen microenvironment. Inhibition of glycolysis with a derivate of 3-bromopyruvate ester (pBr-PE) sensitized chemotherapy-resistant glioblastoma stem-like cells to standard therapeutic agents and counteracted tumor growth in vivo [194]. Accordingly, the activation of an EMT program repressed fructose-1,6-bisphosphatase through a Snail-G9a-Dnmt1 complex causing the production of ROS and enhancement of glycolysis under hypoxia in breast CSCs [15].

Cancer metabolic reprogramming is indeed influenced by proximity to blood vessels and thus the availability of oxygen. By using a methodology capable of sorting of stroma-free glioblastoma cells and based on the distance of cells from vasculature, Kumar et al. demonstrated that glioblastoma cells positioned in close proximity to blood vessels were highly chemo- and radio-resistant and possessed an OXPHOS metabolism [195]. These observations shed light on the presence of metabolic zonation, which means that cancer cell metabolism is highly heterogeneous within the same tumor due to the presence of different metabolic niches that favor the expansion of CSCs [105,196].

Besides being directly involved in nutrients exchange, CAFs secreted cytokines are involved in the activation of EMT, which is associated with metabolic reprogramming of CSCs [197]. In particular, several studies demonstrated that EMT prompts glycolytic flux in CSCs, at the expense of mitochondrial respiration [3]. Interestingly, glucose consumption by cancer cells can in turn can suppress the immune response [198] and create an acidic microenvironment that favors cancer progression [199].

Beyond the cytokine-driven effects, cancer and TME cells can communicate and benefit from the exchange of vesicles-contained messengers. Exosomes are nanovesicles (40–100 nm in diameter) that are released from most cell types into the extracellular space after fusion with the plasma membrane [200,201]. Many reports have highlighted that exosomes play important roles in immune response and tumor progression [200]. It was demonstrated that cancer cells secrete a larger number of exosomes than normal cells. Tumor-derived exosomes are necessary for cell-to-cell communication. They transport a cargo of growth factors, chemokines, miRNAs and other small molecules [202,203]. Moreover, it has been demonstrated in multiple cancer models that the exposure of cancer cells to the conditioned media of CAFs promotes cancer cell growth. Besides the essential role played by CAFs in secreting factors that shape TME to allow the expansion of cancer cell population, CAFs secrete micro vesicles that act as cargo for nutrients, including intermediate metabolites, lipids and amino acids, which are ultimately absorbed by cancer cells [204]. Microvesicles released in the TME by stromal cells and loaded with several metabolites, following the uptake by cancer cells, directly promote cell proliferation and fuel tumor progression [204].

Moreover, CAF-derived exosomes loaded with miRNAs impact metabolism, migration, invasion and metastasis in CSCs. Exosomal miR-21, miR-378e and miR-143, horizontally transferred from CAFs to breast CSC, encouraged the formation of mammospheres concomitantly with the acquisition EMT and stemness markers [205].

Interestingly, in metastatic breast cancer, extracellular vesicles released by CAFs contain mitochondrial DNA that is internalized by CSCs and used to sustain OXPHOS and exit from quiescence induced by hormonal therapy [206].

## 7. Concluding Remarks and Future Perspectives

Herein we show an overview on the metabolic demand of CSCs highlighting the importance of looking at the ‘whole picture’ of tumor metabolism. CSCs possess a heterogeneous metabolism that can vary in different zones of the tumor in response to nutrients and oxygen supply.

Novel frontiers of cancer treatment are focused on the use of biocompatible particles, usually conjugated with anticancer compounds, which are capable of releasing the drug directly at the tumor site. For instance, gold nanoparticles conjugated with salinomycin, caused negligible damage to healthy cells while inducing cell death of the subpopulation of CD44^+^/CD24^−^ breast CSCs by inducing ferroptosis, consisting in the accumulation of iron and ROS derived from lipid oxidation [207].

Thus, the identification of metabolic traits of CSCs and the adaptation of CSCs metabolism in response to a hostile or supportive TME led to the development of promising therapeutic treatments, summarized in Table 2, which interfere with CSCs metabolism and thus, disease progression.

## Figures and Tables

**Figure 1 cancers-12-01436-f001:**
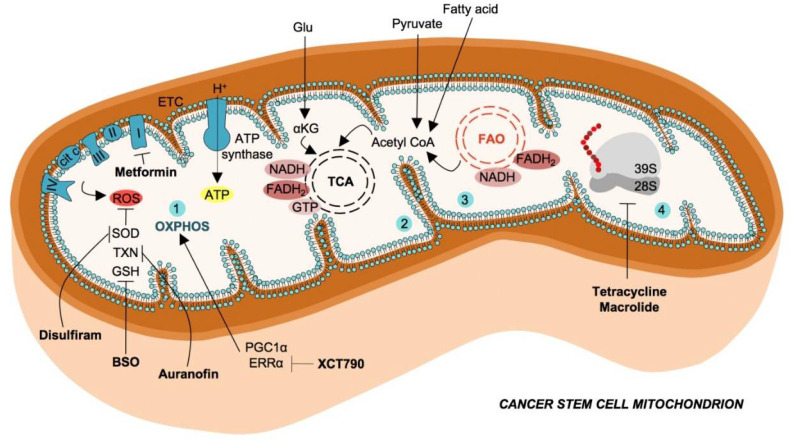
Mitochondrial metabolism in cancer stem cells and therapeutic strategies.Cancer stem cells exhibit enhanced catabolism of glutamate, pyruvate and fatty acids (FAs) to generate intermediate metabolites, which converge in tricarboxylic acid (TCA) cycle. The oxidative phosphorylation (OXPHOS) (1), mediated by the electron transport chain (ETC), use the reducing equivalents (FADH_2_ and NADH) produced by tricarboxylic acid (TCA) cycle (2) and fatty acids oxidation (FAO) (3), with the scope of producing ATP molecules. Mitochondrial ETC is also the main endogenous source of reactive oxygen species (ROS), which possess several scavenger enzymes and molecules in CSCs such as superoxide dismutase (SOD), thioredoxin (TXN) and glutathione (GSH). These antioxidant mechanisms are targeted by Disulfiram, auranofin and L-buthionine-sulfoximine (BSO), respectively. Cancer stem cells possess a high number of mitochondria, mitochondrial ribosomes, necessary for protein synthesis are targetable with antibiotics such as tetracyclines and macrolides (4).

**Figure 2 cancers-12-01436-f002:**
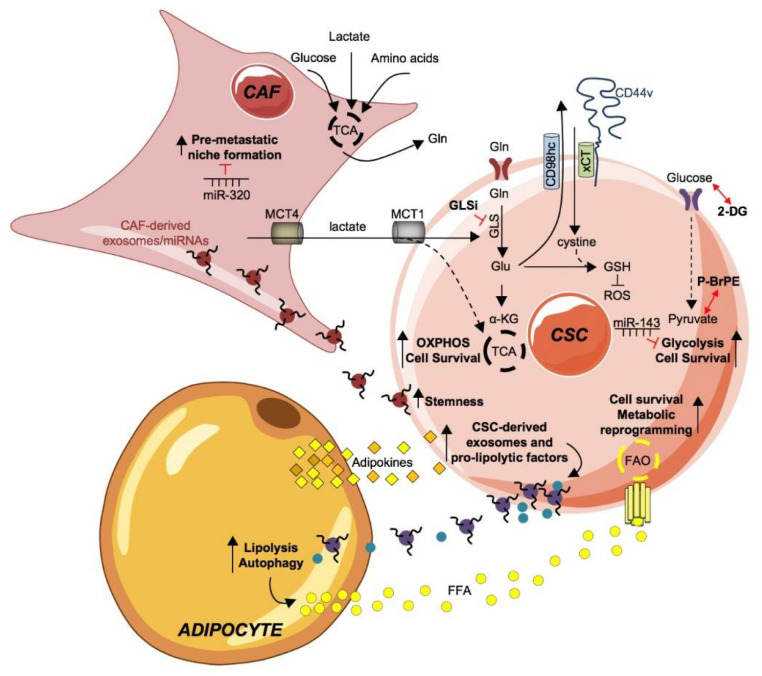
Metabolic fingerprint of cancer stem cells and their coupling with tumor microenvironment. The tumor microenvironment and every single part of it is an important element to be taken into consideration in tumor development. Through the production and excretion by the cancer associated fibroblast (CAF) of intermediate metabolites, such as lactate and glutamine (Gln), they are able to meet the continuous and demanding energy needs of CSCs. Gln is internalized by its transporter ASCT2 and converted in glutamate (Glu), by the enzyme glutaminase (GLS), finally inducing OXPHOS and cell survival of CSCs. GLS activity is blocked by GLU inhibitors. Glu contributes to the formation of reactive oxygen species (ROS) inhibitor molecule glutathione (GSH) by acting as an exchange ion, through the CD44v-xCT transporter, for the import of cystine, which is necessary for the production of GSH together with Glu. The uptake of glucose through the GLUT foster a glycolytic phenotype in CSCs boosting cell survival. The administration of the glycolysis inhibitors 2-deoxyglucose (2-DG), 3-bromopyruvate ester (pBr-PE) or miR-143 proved to effective in dampening the expansion of the CSC subpopulation.The crosstalk of CAF and CSC is possible through CAF-derived exosomes/miRNAs, including miR-21, miR-378e and miR-143. CAF-derived exosomes/miRNAs are internalized by CSC, sustaining stemness and positively influences tumor progression. Conversely, miR-320 counteracts pre-metastatic niche formation by CAFs.Adipocyte-secreted adipokines are taken upby CSC and stimulate the production and release of exosomes and pro-lipolytic factors, including miRNA-144 and miRNA-126, which in turn active lipolysis and autophagy pathways in adipocytes. Consequently, free fatty acids (FFA) are released in the tumor microenvironment (TME) that, upon absorption by CSCs through the fatty acid transporters (fatty acid translocase, FAT/CD36; fatty acid transport protein, FATP and fatty acid-binding protein, FABP), undergo fatty acid oxidation (FAO) to sustain CSCs expansion and metabolic reprogramming.

**Table 1 cancers-12-01436-t001:** Metabolic phenotypes of cancer stem cells in various cancer models.

Cancer Type	Study Setting	Tumor Cells	Methods	Metabolic Phenotype	Reference
Lung cancer	In vitro In vivo	Tumorspheres	Clark-type oxygen electrode Glucose consumption determination	OXPHOS	[20]
Glioblastoma	In vitro	Tumorspheres	Seahorse Citrate determination	OXPHOS	[21]
Non-small cell lung cancer	In vitro In vivo	Tumorspheres	UPLC/MS	GLYCOLYSIS	[22]
Breast cancer	In vitro In vivo	CD44^+^ CD24^low^ EPCAM+ cells	Seahorse	GLYCOLYSIS	[15]
Pancreatic cancer	In vitro In vivo	CD133^+^ CD44^high^ cells	Seahorse	OXPHOS	[23]
Hepatocellular carcinoma	In vitro In vivo	CD133+ cells	GC-MS	GLYCOLYSIS	[16]
Pancreatic cancer	In vitro In vivo	CD133^+^ cells	Seahorse RNA seq	OXPHOS	[24]
Head and neck squamous cell carcinoma	In vitro In vivo	Tumorspheres	UPLC-MS/MS	GLYCOLYSIS	[25]
Acute Myeloid Leukemia	In vitro In vivo	CD34^+^ cells	Seahorse UHPLC-MS	OXPHOS	[26]
Breast cancer	In vitro In vivo	Tumorspheres	RNA seq NADH determination	INTERMEDIATE	[27]
Colorectal cancer	In vivo	Cancer intestinal stem cells	LC-MS	GLYCOLYSIS	[14]

**Table 2 cancers-12-01436-t002:** Metabolic Therapeutic Targets of Cancer Stem Cells.

Cancer Type	Target Candidate	Drug	References
Breast cancer Pancreatic ductal adenocarcinoma	Mitochondria	Metformin	[24,49]
Glioblastoma	Glycolysis	3-bromopyruvate ester	[194]
Breast cancer Glioma Pancreatic cancer	Fatty acid synthase	Cerulenin Cerulenin C75	[86] [83] [85]
Glioblastoma	Hexokinase 2	Mir-143	[154]
Breast cancer	Estrogen-related Receptor α	XCT790	[48]
Multiple tumor types	Mitochondrial ribosomes	Macrolide	[52]
Multiple tumor types Leukemia	Mitochondrial ribosomes	Tetracycline	[52,53]
Hepatocellular cancer	Pyruvate dehydrogenase Kinase 4	Mir-122	[160]
Hepatocellular cancer	Inducer of ROS	Paraquat	[16]
Colon cancer Glioblastoma	Stearoyl-CoA desaturase-1	Betulinic acid CAY10566	[139,141]
Colon cancer	Iron-sulfur cluster assembly protein	Mir-210	[161]
Breast cancer	Aldehyde dehydrogenase	Disulfiram	[62]
Breast cancer	Stearoyl-CoA desaturase-1	Mir-600	[157]
Breast cancer	Thioredoxin	auranofin	[27]
Breast cancer	Glycolysis	2-deoxyglucose	[27]
Breast cancer	Glutathione	L-buthionine-sulfoximine	[27]
Colorectal cancer	Mitochondria and GLS1	Metformin and Glutaminase inhibitor (GLSi)	[75]
Breast cancer	3-hydroxy-3-methyl-glutaryl-coenzyme A reductase	Simvastatin	[133]
Breast cancer	Mitochondria	Salinomycin	[207]

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
