# Peer review of "Metabolic Escape Routes of Cancer Stem Cells and Therapeutic Opportunities"

_cancers, 2020, doi:10.3390/cancers12061436_

Round 1
Reviewer 1 Report
The review is well designed and written well.
The problems in the cancer stem cells metabolism are well described.
Small editorial change need to be done:
line 93: transition of mesenchymal breast CSCs harboring low levels of ROS toward a ROShigh phenotype of" - space between ROS and high
The little concern is absence the reference for clinical trials.
line 558
"For example, a mimic of miR-34 is used like a tumor suppressor in Phase I clinical trials. It regulates lactate dehydrogenase A."
Author Response
We thank the reviewer for the appreciation to the review content.
As suggested by this reviewer we carefully revised and edited the manuscript.
Missing clinical trial identifiers have now been included thoroughly the manuscript (NCT01266486; NCT02755844; NCT01620593; NCT02978547; NCT01829971; NCT01200420; NCT01872936; NCT02031133; NCT02508090; NCT01829971; NCT03713320).
Reviewer 2 Report
The review extensively collected all the recent literature on metabolic routes of cancer cells, CSC and role of Tumor micro environment in CSC-metabolism.
However, I have concerns about the review where some topic is too focused on cancer cells rather focused on CSC metabolism. The metabolic therapeutic targets of CSC is not emphasised enough in the article. The article needs to subdivided into more subtopic to make more visible and readable the information on the review. Below are my main concerns that needs to be addressed before acceptable to Journal.
1) Authors focused on Mitostemness is important part of CSC- regenerative capacity is commendable. I am wondering what's relevant of line 175-178?
2) Line 230-268 needs to be concise, it's too lengthy introduction to FA beyond the focus of CSC-metabolic routes? Does FA de novo synthesis is specific to CSC?
3) I suggest authors to remove line 265-268 that's not required here?
4) FA-aberrant metabolism discussed in the article ( line 365-411) too focused on cancer cells though title focused on CSC. Either authors edit information more relevant to CSC or title needs to be changed to cancer cells instead of CSC?
5) I would recommend authors to include tables that contains information on various metabolites observed in the CSC of various cancers, model ( in vivo /in vitro) used in the study and phenotypes from the literature.
6) I strongly feel metabolic therapeutic target of CSC discussed very less in the manuscript. I suggest authors should include a subtopic to discuss recent observation on this or a table containing promising candidate of metabolic pathways that are targetable for therapeutic purpose.
Author Response
We thank the reviewer for the constructive comments that will ameliorate the quality of the manuscript.
The reviewer pointed out that some topics should focus more on CSC and to deeper discuss the most promising anti-cancer treatments aiming at targeting cancer metabolism. Here below our reply to all reviewer concerns:
1) As suggested by this reviewer we modified line 175-178.
2) We shortened the introduction regarding FA metabolism. We are now focusing more on the FA de novo synthesis in CSCs by citing the not abundant studies, as pointed out by this reviewer, present in literature (Yasumoto, Y.; PLoS One 2016; Pandey, P.R. Oncogene 2013; Brandi, J. J Proteomics 2017; Wang, X. Oncogene 2013).
3) We removed line 265 to 268.
4) As suggested by this reviewer we changed the title of the paragraph as we refer now to cancer cell in general. We made the general description of FA-aberrant in cancer cells metabolism more concise while focusing more on CSCs.
5) In the revised version of the manuscript we added a new table (Table 1), describing the two major phenotype of CSCs, glycolytic and oxidative, in various cancer models.
6) In order to emphasize the role and the importance of anti-cancer metabolic drugs on CSCs we included a new table (Table 2) with detailed information on drugs and their targets in various cancer types. We are also including new information regarding metabolism-targeting drugs in CSCs (Zhao, Y. 2019; Ehmesen, S. 2019; Pinkham, K. 2019; Wang, X. 2013; Yasumoto, Y. 2016; Brandi, J. 2017; Iliopoulos, D. 2011).
Round 2
Reviewer 2 Report
The authors addressed all my previous concerns in the revised version of the manuscript. This review will be a resource for researchers in cancer metabolism and cancer stem cell field. I recommend the current manuscript for the journal.